# A Cohort Study Investigating Zearalenone Concentrations and Selected Steroid Levels in Patients with Sigmoid Colorectal Cancer or Colorectal Cancer

**DOI:** 10.3390/toxins16010015

**Published:** 2023-12-27

**Authors:** Sylwia Lisieska-Żołnierczyk, Magdalena Gajęcka, Michał Dąbrowski, Łukasz Zielonka, Maciej T. Gajęcki

**Affiliations:** 1Independent Public Health Care Centre of the Ministry of the Interior and Administration, and the Warmia and Mazury Oncology Centre in Olsztyn, Wojska Polskiego 37, 10-228 Olsztyn, Poland; lisieska@wp.pl; 2Department of Veterinary Prevention and Feed Hygiene, Faculty of Veterinary Medicine, University of Warmia and Mazury in Olsztyn, Oczapowskiego 13, 10-718 Olsztyn, Poland; michal.dabrowski@uwm.edu.pl (M.D.); lukasz.zielonka@uwm.edu.pl (Ł.Z.); gajecki@uwm.edu.pl (M.T.G.)

**Keywords:** cohort study, zearalenone, estradiol, progesterone, sigmoid colorectal cancer, colorectal cancer

## Abstract

The aim: In this study was to determine if sigmoid colorectal cancer (SCC) and colorectal cancer (CRC) in women (W) and men (M) is accompanied by zearalenone (ZEN) mycotoxicosis and changes in selected steroid levels. Materials and Methods: This cohort study was conducted on female and male patients selected from a population based on the presence of SCC or CRC, which was accompanied by the presence or absence (control group) of ZEN in their blood. The control group consisted of 17 patients with symptoms of SCC and CRC, where ZEN and its metabolites were not detected in the peripheral blood. The experimental groups comprised a total of 16 patients with SCC and CRC, where ZEN, but not its metabolites, was detected in their peripheral blood samples. Results: In groups SCC and CRC, the ZEN levels were very high, in the range from 214 to 289 ng/mL of blood. Considerable variations were observed in the concentrations of steroid hormones. Estradiol (E_2_) levels ranged from 166.25 (group C) to 325 pg/mL (group CRC) in women and from 98 (group C) to 95.5 pg/mL (group CRC) in men. Progesterone (P_4_) levels ranged from 12.09 (group C) to 13.64 ng/mL (group SCC) in women and from 6.98 (group CRC) to 12.01 ng/mL (group C) in men. Conclusions: These results indicate that post-menopausal women and similarly aged elderly men have a high and individualized demand for estrogen that is relatively effectively met by ZEN, which triggers qualitative changes in estrogen receptors. The shortage of ZEN metabolites (values under the sensitivity of the method) confirmed the high estrogen demand in the studied subjects. The presence or absence of ZEN could have influenced the therapeutic outcomes in the analyzed patients.

## 1. Introduction

Food and feed products are widely contaminated with mycotoxins, which poses a health risk for humans and animals and has serious implications for global commerce. Food and feed can become contaminated during the entire production process, both before and after harvesting due to inadequate handling of raw materials, as well as during storage and distribution. Mycotoxins ingested with food and feed can exert teratogenic, carcinogenic, estrogenic, and immunosuppressive effects in humans and animals, which is why the health risks associated with ZEN and/or its metabolites should be investigated [1].

Zearalenone is a non-steroidal estrogenic mycotoxin described chemically as 6-[10-hydroxy-6-oxo-trans-1-undecenyl]-β-resorcyclic acid lactone. Polyketide synthase genes are involved in ZEN biosynthesis in mold fungi of the genus *Fusarium*, such as *F. graminearum*, *F. culmorum*, and *F. cerealis.* Zearalenone is produced during a series of reactions catalyzed by numerous multienzyme protein complexes formed via polyketide synthases [1].

There is growing evidence to indicate that ZEN, as a mycotoxin [2] and endocrine-disrupting chemical (EDC) [3] which affects tissues containing estrogen receptors (ERs) [4] and the nervous system [5], may disrupt hormonal activity, including in humans [6]. This *Fusarium* toxin is classified as a xenoestrogen due to its structural similarity to β-estradiol and affinity for ERs [7], which contributes to the development and progression of cancer. Zearalenone causes hormonal and reproductive dysfunctions, in particular in pigs. It also affects digestive [6,8], immune [8,9], and nervous systems [10].

The hormonal system, also referred to as the endocrine system, regulates bodily functions and is responsible for the maintenance of homeostasis [3]. It consists of a complex mixture of chemical substances that interact with each other and with exogenous substances. Hormonal dysfunctions can lead to serious diseases and disorders that impair bodily functions. Endocrine disorders can be caused by numerous factors [10], including undesirable substances such as EDCs, e.g., ZEN [11,12,13,14,15]. These compounds disrupt various processes in the endocrine system, including biosynthesis, transport, metabolism, receptor and post-receptor expression, and feedback mechanisms.

The authors’ previous studies have shown that ZEN and its metabolites exert specific effects on the intestines [16] in periods of physiological estrogen deficiency (Figure 1). These effects are most pronounced in prepubertal individuals, but also in post-menopausal women and similarly aged men. In a hypoestrogenic state, the effectiveness of cell proliferation and differentiation processes decreases, which can increase the permeability of the intestinal mucosa and contribute to lymphocyte recruitment and chronic inflammatory infiltration [17]. It is highly probable that the expansion of malignant tumors such as SCC and CRC is preceded by the translocation of live bacteria from the gastrointestinal tract across the epithelial mucosa to the lamina propria, mesenteric lymph nodes [18], and other organs [19] as part of the intestinal response to chronic systemic hypoestrogenism. In the current situation, the presence of ZEN and its metabolites in plant materials may contribute to hyperestrogenism, or, rather, this condition should be referred to as supraphysiological hormonal levels [20], as it contributes to an increase in metabolic pro-cases which are not always controlled but may also deepen the processes of intestinal inflammation [21]. It should also be borne in mind that, in systematic reviews and meta-analyses of various human populations, opinions are divided on the effect of steroid hormones on the risk of CRC. Some believe that the levels of individual steroid hormones are irrelevant [22,23], while others believe that their presence increases the risk of colorectal cancer [24,25]. This ambiguity has led to the hypothesis that the exogenous mycoestrogen ZEN may play a specific role in the etiopathogenesis of neoplastic lesions in the distal colon in the elderly. 

The arguments presented above became the basis for a cohort study of patients with SCC and CRC in order to check whether they had been exposed to ZEN and/or its metabolites prior to admission to the hospital and whether there were changes in the levels of selected steroid hormones (estradiol—E_2_ and progesterone—P_4_) in their peripheral blood.

## 2. Results

The presented results provide valuable inputs for assessing the impact of relatively high, natural doses of ZEN on carcinogenesis processes in the end segments of the colon in women (W) and men (M). A specific interaction was observed between ZEN concentration and the levels of selected steroids (E_2_ and P_4_) in the peripheral blood of women and men diagnosed with SCC and CRC. A preliminary data analysis documents a set of highly individualized results for the parent substance and the absence of selected ZEN metabolites (alpha-zearalenol—α-ZEL; beta-zearalenol—β-ZEL) in the studied subjects.

### 2.1. Clinical Aspects

Clinical aspects of zearalenone mycotoxicosis were not noted during the patients’ hospitalization.

### 2.2. Zearalenone Concentrations in Peripheral Blood

In patients from groups SCC and CRC, the ZEN concentrations in the peripheral blood were nearly 100 times higher than those noted in omnivorous animals, including in prepubertal gilts exposed to ZEN in quantities of 5, 10, and 15 µg/kg of body weight over a period of 42 days [11].

No significant differences in ZEN levels were observed between W and M from groups SCC and CRC (Figure 2). The only exception were women with symptoms of CRC, where ZEN concentrations were meaningfully higher (*p* ≤ 0.05) than in women with symptoms of SCC (difference of 64.95 ng/mL) and in men with symptoms of SCC (difference of 60.02 ng/mL) and CRC (difference of 74.53 ng/mL). In group C, ZEN was not detected in patients with SCC or CRC (which implies that carcinogenesis was not accompanied by ZEN mycotoxicosis), possibly because these patients consumed plant-based foods that were free of ZEN. Zearalenone metabolites (α-ZEL and β-ZEL) were not detected in all groups (values below the sensitivity of the method).

### 2.3. Estradiol and Progesterone Concentrations in Blood

The mean age of the studied patients was 60.53 years for women and 69.66 years for men (Table 1). The mean age of patients with SCC, CRC, and ZEN mycotoxicosis was 60.05 years for women (post-menopausal) and 70.35 years for men.

The presented data indicate (Table 1) that women were 9.13 years younger on average than men. The female patients in the control group (mean age—61.5 years) were 1.45 years older on average than the female patients in groups SCC and CRC (mean age—60.05 years). Estrogen levels decline after menopause (to 11.2–42.0 pg/mL), but, despite the fact that all the studied female patients were post-menopausal, their estrogen levels did not decrease. The male patients in the control group (mean age—68.3 years) were 2.05 years younger on average than the male patients in groups SCC and CRC (mean age—70.35 years).

### 2.4. Statistical Comparison of the Values of Both Tested Hormones in Women in All Experimental Groups

In the women, the mean blood levels of E_2_ ranged from 166.25 pg/mL in group C to 286.6 pg/mL in group SCC and 325.0 pg/mL in group CRC (Figure 3). Despite considerable variation in the mean E_2_ levels, meaningful differences were not observed between the groups. The studied parameter was 61.5% and 95.5% higher in groups SCC and CRC, respectively, than in the control group. The absence of significant differences resulted from a high standard deviation, which reached 130.4 pg/mL in group C, 336.9 pg/mL in group SCC, and 371.3 pg/mL in group CRC.

No statistical differences in P_4_ concentrations were found between groups. The mean values of this parameter ranged from 12.09 ng/mL in group C to 13.64 ng/mL in group SCC and 12.64 ng/mL in group CRC. The standard deviation was specified at 3.81 ng/mL in group C, 6.42 ng/mL in group SCC, and 4.84 ng/mL in group CRC.

### 2.5. Statistical Comparison of the Values of Both Tested Hormones in Men in All Experimental Groups

In the male patients, the mean E_2_ levels in the peripheral blood samples ranged from 95.25 pg/mL in group SCC to 95.50 pg/mL in group CRC and 98.00 pg/mL in group C (normal range: 11.2 to 50.4 pg/mL) (Figure 4). It should be noted that the mean concentrations of E_2_ were several times lower in men than in women (Figure 3). The presence of ZEN in plant-based foods induced a non-significant decrease in the mean blood levels of E_2_ in groups SCC and CRC (by 2.75 and 2.50 pg/mL, respectively) relative to group C.

In the men, the mean P_4_ levels in the peripheral blood samples (Figure 4) differed significantly (by 4.50 ng/mL; * *p* ≤ 0.05) between group SCC (7.51 ng/mL) and group C (12.01 ng/mL).

### 2.6. Statistical Comparison of the Values of Both Tested Hormones between Women and Men in All the Groups in the Experiment

A comparison of the mean E_2_ levels in the female and male patients (Figure 5) revealed significant differences (173.35 pg/mL; * *p* ≤ 0.05) only in group SCC. High but non-significant differences in the mean E_2_ levels were noted between the men and the women in the remaining groups (229.5 pg/mL in group CRC, and 68.25 pg/mL in group C). The absence of significant differences resulted from a high standard deviation, which reached 120.4 in group C and 365.5 in group CRC.

Highly significant differences (** *p* ≤ 0.01) in the medium levels of P4 (Figure 5) were noted between W and M patients in group SCC. The progesterone levels were 6.13 ng/mL (44.95%) lower in the men than in the women. Significant differences between genders were not observed in the remaining groups, despite the fact that the mean levels of P_4_ differed considerably in group CRC (by 5.66 ng/mL, i.e., 44.47%). In group C, the difference between genders was determined at only 0.08 ng/mL (0.66%). The absence of significant differences resulted from a high standard deviation, which reached 5.75 ng/mL in group CRC.

## 3. Discussion

Zearalenone was detected in a large number of blood samples, which confirms the research hypothesis that this mycotoxin is ubiquitous and that ZEN influences the concentrations of two important steroid hormones, E_2_ and P_4_.

### 3.1. Zearalenone

The effects of the present study suggested that the presence of ZEN and its metabolites in the peripheral blood of women and men with symptoms of SCC and CRC can be attributed to assimilation and biotransformation processes. Zearalenone concentrations differed across the studied subjects, but significant differences were rarely noted, although SD values varied considerably (Figure 1). Interestingly, two ZEN metabolites, α-ZEL and β-ZEL, were not detected in any of the analyzed blood samples (values under the sensitivity of the method). The above could be attributed to the fact that post-menopausal women have an increased demand for estrogen [26], and ZEN is a compound with estrogen-like activity [11,27]. A physiological decline as well as rapid and irregular fluctuations in the absolute and relative levels of estrogens and P_4_ are noted in post-menopausal women (Table 1). Therefore, the presence of ZEN (a substance with high estrogenic activity) [28] could exert similar effects to hormone therapy by promoting physiological functions [29] and enhancing metabolic processes but (unfortunately) in an uncontrolled manner. These observations suggest that prolonged exposure to ZEN (which is also referred to as a mycoestrogen, a xenoestrogen, or an undesirable substance) is associated with tumors not only in the reproductive system [3] but also in the gastrointestinal tract [6,12,30]. According to Cieplińska et al. [13] and Ruan et al. [31], the processes observed in the gastrointestinal tract result from new mechanisms underlying mycotoxin-induced toxicity, which determine the quantitative and qualitative composition of key gut microbiota.

Zearalenone is also metabolized in the liver, and liver damage is the most common effect of ZEN’s toxicity [32]. According to Llorens et al. [33], Liu et al. [34], and Damiano et al. [35], ZEN is capable of generating reactive oxygen species (ROS) in a dose-dependent manner as well as by changing the expression levels of the antioxidant enzymes and genes associated with oxidative stress both in vivo and in vitro. Oxidative stress [36] is caused by an imbalance in the production of ROS and the body’s ability to detoxify these reactive products or repair the damage that has occurred. Exposure to large doses of ZEN (Figure 2) produces hepatotoxic effects and decreases the effectiveness of detoxification processes [37], which could explain high ZEN concentrations in the blood of the studied patients, particularly in the women.

It should also be noted that ZEN is a substrate that regulates the expression of genes encoding hydroxysteroid dehydrogenases (HSDs), which act as molecular switches and allow for the pre-receptor modulation of steroid hormones [38]. At the same time, endogenous estrogens and exogenous xenobiotics, such as ZEN, influence the expression of ERs that regulate proliferation and apoptosis in enterocytes, in particular in the distal gut [4].

These processes, alone or in combination, could have influenced the concentrations of ZEN and its metabolites in the peripheral blood of patients from both study groups. Zearalenone is a compound with multiple biological activities, and it can inhibit the synthesis and secretion of the follicle-stimulating hormone (FSH) [10] due to negative feedback, which decreases steroid production and contributes to hypogonadism, including in males [39].

### 3.2. Estradiol

Post-menopause is the last stage of the menopausal transition period, and it marks the end of the reproductive stage of life (Table 1). During this period, complex biological changes are observed in both sexes (study groups). The gradual decline in the endocrine functions of the ovaries can lead to physiological disorders, including tumors in the distal gut [6,16]. The estrogenic activity of ZEN has been widely demonstrated in females of many animal species [4,10] as well as in women [29]. Similarly, observations were made in this study according to which E2 concentration were significant but not significantly higher in the women in both study groups (by 61.5% in group SCC and by 95.5% in group CRC) than in the control group (Figure 3). As a result, hyperestrogenism or supraphysiological estrogen levels [20] were observed in these patients, induced by the reaction to the appearance of ZEN, an exogenous mycoestrogen. “Free” ZEN that is not utilized by the macroorganism is captured by ERs in the gastrointestinal tract [4] and induces qualitative changes by enhancing the expression of ERs and increasing the concentration of E_2_. The zearalenone levels were very high in the examined patients (more than 100 times higher than in pigs exposed to the lowest observed adverse effect level (LOAEL) dose of ZEN [4]), which triggered changes in the expression of ERs, in particular ER*ß*, in the descending colon, promoted the growth of gut microbiota, and increased ZEN’s genotoxicity [13,24,25,40]. These observations indicate that the consumption of food products contaminated with ZEN can intensify steroidogenesis processes in post-menopausal women, e.g., by changing the expression of HSDs.

In the male patients, the mean blood levels of E_2_ did not differ between groups SCC, CRC, and C (where ZEN was not detected) (Figure 4), and they were two or even three times lower than in the women. These results are very difficult to comment on due to the general scarcity of research studies on the subject. Therefore, attempts were made to extrapolate the results. Mycoestrogen should interact with ERs and mediate estrogenic responses. It should be noted that the interactions between nervous [5], hormonal, and immune systems are accountable for keeping homeostasis at the cellular and systemic levels. Any disruption in the communication between homeostatic systems, which are essential for healthy functioning, can speed up the progression of disease [5]. Very few studies have investigated ZEN’s influence on the immune system and its contribution to the development of autoimmune diseases [18] already at the gut level. Zearalenone inhibits the neuroendocrine activity of the hypothalamus, which controls the release of hormones from the pituitary gland and affects the functions of endocrine glands [10]. As a result, homeostatic systems may be unable to control metabolic processes [41], including in the distal segments of the gastrointestinal tract.

### 3.3. Progesterone

The progesterone levels were very high in the blood of W and M patients who had been exposed to unknown quantities of ZEN in naturally contaminated food. The reference levels of P_4_ are <4.0 ng/mL for post-menopausal women (mean age of 60.5 years) and 0.2–1.38 ng/mL for elderly men (mean age of 69.6 years) (Table 1). The progesterone levels also exceeded the reference range in the C group (Figure 5), where patients were diagnosed with the same diseases, but ZEN was not detected in their blood samples. The values noted in group C could indicate that the absence of ZEN did not induce significant changes in the P_4_ levels in both sexes (post-menopausal women and similarly aged men). Very high concentrations of both endogenous steroid hormones (Figure 5) indicate, similarly to a previous study of prepubertal gilts [11], that the strength of a linear relationship points to a positive correlation between ZEN vs. E_2_ and P_4_ concentrations. The presence of a positive correlation between ZEN and P_4_ stimulated the production of antibodies that are not asymmetrically glycosylated and are able to trigger immune effector mechanisms [42].

Somewhat different conclusions can be drawn based on a detailed analysis of P_4_ levels in both sexes. In women, ZEN induced a considerable, although non-significant, increase in P_4_ levels relative to group C (Figure 3). In men, ZEN exerted an opposite, suppressive effect in men, where it decreased the P_4_ levels in groups SCC (significant difference) and CRC relative to group C (Figure 4). This is a very important consideration in autoimmune diseases where autoimmunization targets autoantigens [18].

### 3.4. Conclusions

The present cohort study was conducted on female and male patients selected from a population based on the presence of malignant tumors (SCC or CRC), which were accompanied by the presence or absence (control group) of ZEN in their peripheral blood. The results of the study confirmed the research assumptions. A preliminary analysis of the data indicates that the ZEN levels varied considerably across the patients, but ZEN metabolites (α-ZEL and β-ZEL) were not detected. The following conclusions can be formulated based on the presented results: (i) ZEN was identified in 48.5% of the analyzed blood samples, and it influenced the concentrations of two very important steroid hormones (E_2_ and P_4_); (ii) ZEN induced specific interactions between the analyzed steroid hormones (by stimulating the production of E_2_ and suppressing the production of P_4_) in the peripheral blood of W and M; and (iii) the presence or absence of ZEN could have influenced therapeutic outcomes in the examined patients. These findings provide valuable inputs for further research on food contamination with ZEN as one of the multiple etiological factors of carcinogenesis in the last segments of the colon in women and men in old age.

## 4. Materials and Methods

### 4.1. Patients

This single-center retrospective study was based on the medical records of female and male subjects with histopathologically confirmed SCC or CRC. The participants were patients of the Independent Public Health Care Center of the Ministry of the Interior and Administration and the Warmia and Mazury Oncology Center in Olsztyn, Poland, and they were recruited for the study over a period of one year from different counties of the voivodeship of Warmia and Mazury. The participants signed informed consent forms, and they were informed that they could refuse access to their personal and medical data at any point during the trial. All the medical records were anonymized.

This cohort study was conducted on female and male patients selected from a population based on the presence of malignant tumors (SCC or CRC), which was accompanied by the inherence or absence (group C) of ZEN in their blood. Changes in the concentration of steroid hormones during natural ZEN mycotoxicosis were determined in the female and male patients. The C group comprised seventeen subjects (51.5% of the study population), including eight W and nine M, whose blood samples were free of ZEN and its metabolites. In group C, eight patients were diagnosed with SCC (four W and four M), and nine patients were diagnosed with CRC (four W and five M). The experimental groups comprised a total of sixteen patients with malignant tumors of the distal gut (eight W and eight M), where ZEN, but not its metabolites, was detected in their peripheral blood samples, including nine patients diagnosed with SCC (five W and four M, 27.3% of the study population) and seven patients diagnosed with CRC (three W and four M, 21.2%).

### 4.2. Blood Sampling

The blood samples to be tested for ZEN, α-ZEL, β-ZEL, E_2_, and P_4_ were taken accordingly to the method described by Guder et al. [43]. The blood was sampled between 7 a.m. and 9 a.m., after a 12 h fasting period and 1 h after the last cigarette, in a relaxed and peaceful environment, before the implementation of diagnostic and therapeutic procedures. The blood was collected with the use of Vacuette Quickshield Safety Tube Holders (Greiner Bio-One GmbH, Kremsmunster, Austria). Blood samples of 20 mL each were collected by venipuncture from the basilic vein or the median cubital vein into EDTA-K2 tubes (Vacuette Tube, Greiner Bio-One GmbH, Kremsmunster, Austria). The collected samples were not stored. The tubes were tightly sealed, kept in an upright position, and protected from light; the samples’ contact with air was limited. Within one hour from collection, the samples were transported to a laboratory where they were centrifuged at 3000 rpm for 20 min at 4 °C (THERMO FISHER SCIENTIFIC; model SORVALL X4R PRO-MD; Rotor FIBERLITE f13-14X50cy; Merck KGaA, Darmstadt, Germany). The separated plasma was stored at −18 °C until each sample was analyzed for the presence of ZEN, α- and β-ZEL, E_2_, and P_4_ concentrations.

### 4.3. Mycotoxin Extraction Procedure

The presence of ZEN, α-ZEL, and β-ZEL in the blood plasma was determined with the use of immunoaffinity columns (Zearala-TestTM Zearalenone Testing System, G1012, VICAM, Watertown, MA, USA). All extraction procedures were conducted in accordance with the manufacturers’ instructions. After extraction, the eluents were placed in a water bath at a temperature of 40 °C and were evaporated in a stream of nitrogen. Next, 0.5 mL of acetonitrile (LCMS grade) was added to the dry residues to dissolve the mycotoxin. The mycotoxin levels were determined using mass spectrometry.

#### 4.3.1. Chromatographic Quantification of ZEN and Its Metabolites

Zearalenone and its metabolites were quantified at the Department of Veterinary Prevention and Feed Hygiene, Faculty of Veterinary Medicine, University of Warmia and Mazury in Olsztyn, Poland. The concentrations of ZEN, α-ZEL, and β-ZEL in the blood plasma were determined with the use of an Agilent chromatographic system (Santa Clara, CA, USA) comprising a G7167A multisampler, a G7104C quaternary pump, a G7116A multicolumn thermostat, and a 6470 double quadrupole LC/TQ mass spectrometer. Chromatographic separation was performed with the use of the ZORBAX Eclipse Plus C18 column (2.1 × 50 mm, 1.8 µm) in a gradient study (Table 2). Mobile phase A was composed of 0.1% formic acid and 0.5 mM ammonium fluoride in water (Merck KGaA, Darmstadt, Germany). Mobile phase B contained 0.1% formic acid and 0.5 mM ammonium fluoride in acetonitrile (Merck KGaA, Darmstadt, Germany). The profile of the gradient elution process is presented in Table 2.

The optimal separation of the analyzed compounds was achieved when the mobile phase flow rate was 0.250 mL/min. The injection volume was 5 µL, and the column temperature was 40 °C. Mass spectrometry with electrospray ionization (Agilent Jet Stream) was conducted in positive ion mode. The following ion source parameters were applied: temperature of nebulizer gas—250 °C; flow rate—8 L/min; nebulizer gas pressure—30 psi; temperature of carrier gas—350 °C; flow rate—12 L/min; and capillary voltage—3300 V.

The optimized parameters in the multiplicity reaction monitoring (MRM) mode and the retention times of the tested analytes are presented in Table 3.

The mycotoxin concentrations were determined with an external standard and expressed in ppb (ng/mL). Matrix-matched calibration standards were applied in the quantification process to eliminate matrix effects that could decrease sensitivity. The calibration standards were dissolved in the matrix samples based on the procedure that was used to prepare the remaining samples. The material for the calibration standards was free of mycotoxins. The limits of detection (LOD—Table 4) for ZEN, α-ZEL, and β-ZEL were determined as the concentration at which the signal-to-noise ratio decreased to three. The limit of quantification (LOQ) was estimated to be ten times the LOD value (Table 4). The concentrations of ZEN, α-ZEL, and β-ZEL were determined in all the tested groups.

##### Mass Spectrometry Conditions

The mass spectrometer worked with the use of an ESI ion source in a positive polarization mode. The linearity was confirmed with a six-point calibration curve. Table 4 presents the obtained parameters in terms of detection limit, quantification, and linearity of the calibration curve.

The method was developed on the basis of an internal validation procedure, which is an abbreviated protocol of FDA (Food and Drug Administration, pursuant to 21 CFR Part 11—Code of Federal Regulations) guidelines. In the above procedure, the LOD, LOQ, and linearity were determined (Table 4). Due to the use of an extraction curve (each point of the curve is an extract from the fortified matrix), the issue of determining the matrix effect and recovery has been omitted. A calibration curve comprising six increasing concentrations of ZEN, α-ZEL, and β-ZEL was developed using five parallel extracts from the fortified matrix for each point of the curve (five analytes per calibration point). The concentration of points was 1, 5, 10, 20, 40, 80 ng/mL of serum. The calibration points, which were also the control points, allowed us to determine the repeatability of the RSD% for ZEN (97.63%), α-ZEL (95.72%), and β-ZEL (96.11%).

#### 4.3.2. Statistical Analysis

The data were processed statistically at the Department of Discrete Mathematics and Theoretical Computer Science, Faculty of Mathematics and Computer Science of the University of Warmia and Mazury in Olsztyn, Poland. The bioavailability of ZEN and its metabolites in blood was analyzed in the control group and in the experimental groups. The results were expressed as means (x^−^) with standard deviation (SD). The following parameters were analyzed: (i) differences in the mean values of the control group and experimental group subjects, and (ii) differences in the average values of the men and women diagnosed with SCC or CRC. In both cases, the differences between the mean values were determined using a one-way ANOVA. If significant differences were noted between the groups, the differences between paired means were determined using Tukey’s multiple comparison test. If all the values were below the LOD (mean and variance equal to zero) in any group, the values in the remaining groups were analyzed using a one-way ANOVA (if the number of the remaining groups was higher than two), and the means in these groups were compared against zero using Student’s t-test. The differences between the groups were determined using Student’s *t*-test. The results were regarded as highly significant at *p* < 0.01 (**) and as significant at 0.01 < *p* < 0.05 (*). The data were processed statistically using the Tibco Statistica v.13.3.0. software (TIBCO Software Inc., Silicon Valley, CA, USA, 2017).

### 4.4. Hormone Levels in Plasma

#### 4.4.1. Estradiol

Estradiol was quantified at the Independent Public Health Care Center of the Ministry of the Interior and Administration and at the Warmia and Mazury Oncology Center in Olsztyn, Poland. The estradiol levels in the plasma were measured in an electrochemiluminescence immunoassay (ECLIA) with the use of Elecsys Progesterone II and Cobas e 411 analyzers (Roche, Basel, Switzerland). The radioimmunoassay (RIA) involved a commercial kit (ESTR-US-CT, producer CIS BIO ASSAYS), and it was based on a previously described procedure [44]. All the measurements were performed in duplicate for each cultured probe. The extraction yield for E_2_ was 90.67 ± 0.73%. The radioactivity of the samples with J125 was measured with an automatic Wallac 1470 WIZARDÆ gamma scintillation counter (Perkin Elmer, Waltham, MA, USA). The one-minute counts were performed with a Geiger counter (with a counting efficiency of 75%). The sensitivity of the assay for E_2_ was 1.36 pg/mL. The standard curve ranged from 2.72 to 550 pg/mL. The intra- and inter-assay coefficients of variation were 5% and 5%, respectively.

#### 4.4.2. Progesterone

Progesterone was quantified at the Independent Public Health Care Center of the Ministry of the Interior and Administration and at the Warmia and Mazury Oncology Center in Olsztyn, Poland. The progesterone levels were determined in an ECLIA electrochemiluminescence assay with the use of Elecsys Progesterone II and Cobas e 411 analyzers (Roche, Basel, Switzerland). In the first stage, the samples were incubated with biotinylated monoclonal antibodies specific for P_4_ and for P_4_ derivatives labeled with a ruthenium complex. The extent to which the hormones were bound to antibodies was determined through their concentrations. Streptavidin-coated microspheres were added in the second stage, and the complex was bound to the solid phase during the interactions between biotin and streptavidin. The quantity of labeled P_4_ bound to the solid phase was inversely proportional to the concentration of P_4_ in the sample. The reaction mix was sucked into a measuring cell where microspheres were magnetically captured on the surface of the electrode. The unbound compounds were removed with ProCell. Voltage was applied to the electrode, and the resulting chemiluminescence was measured with a photomultiplier. The results were read from a two-point calibration curve and a standard curve developed with a barcode verifier. The analytical range of the method was determined by the lower limit of detection and the highest point on the calibration curve at 0.03–60 ng/mL for P_4_. All the analyses were performed in accordance with the manufacturer’s instructions.

#### 4.4.3. Statistical Analysis

The data were processed statistically at the Department of Discrete Mathematics and Theoretical Computer Science, Faculty of Mathematics and Computer Science of the University of Warmia and Mazury in Olsztyn, Poland. The hormone concentrations were measured in two experimental groups and in the control group. The results were expressed as mean values (x^−^) and standard deviation (SD) for each sample. The following assays were performed for every hormone: (i) the differences between the mean values in the experimental groups and the control group were determined, and (ii) the differences between the mean values in the men and women with SCC or CRC were determined. In both cases, the differences were determined using a one-way ANOVA. If the differences between the group means were statistically significant, the differences between pairs of means were determined with Tukey’s multiple comparison test. The equality of variances in the compared groups was evaluated with Levene’s test and the Brown–Forsythe test. If the equal variance hypothesis was rejected in both tests, the significance of the differences was evaluated in the Kruskal–Wallis non-parametric test. In each analysis, the tested values were regarded as highly significant at *p* < 0.01 (**) and as significant at 0.01 < *p* < 0.05 (*). The results were processed in Tibco Statistica v. 13.3.0. (TIBCO Software Inc., Silicon Valley, CA, USA, 2017).

## Figures and Tables

**Figure 1 toxins-16-00015-f001:**
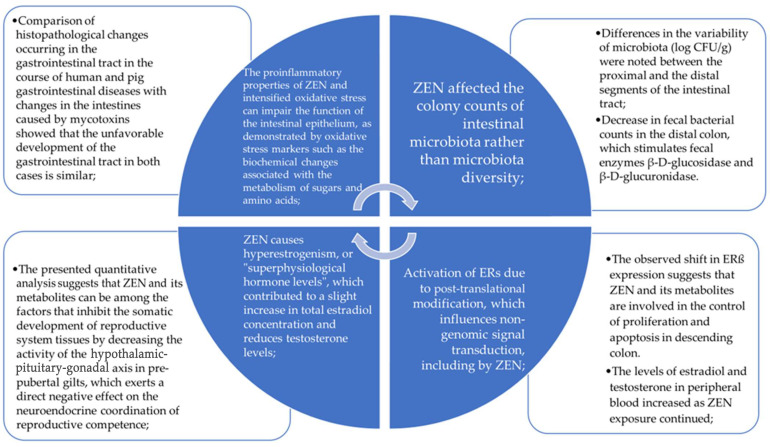
The team’s achievements provide the basis for the hypothesis [4,10,12,13,14,15]—the involvement of ZEN in the etiopathogenesis of colorectal neoplastic lesions.

**Figure 2 toxins-16-00015-f002:**
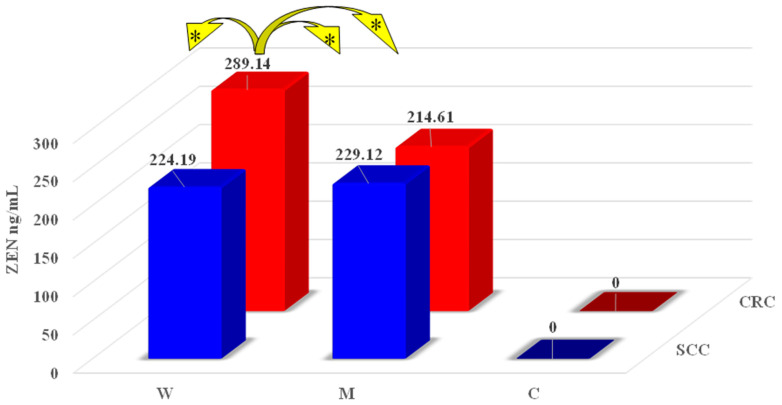
Mean zearalenone (ZEN) concentrations (ng/mL) in the peripheral blood of women and men in groups SCC (sigmoid colorectal cancer) and CRC (colorectal cancer). The limit of detection in group C was set to 0. The differences were considered statistically significant at * *p* ≤ 0.05.

**Figure 3 toxins-16-00015-f003:**
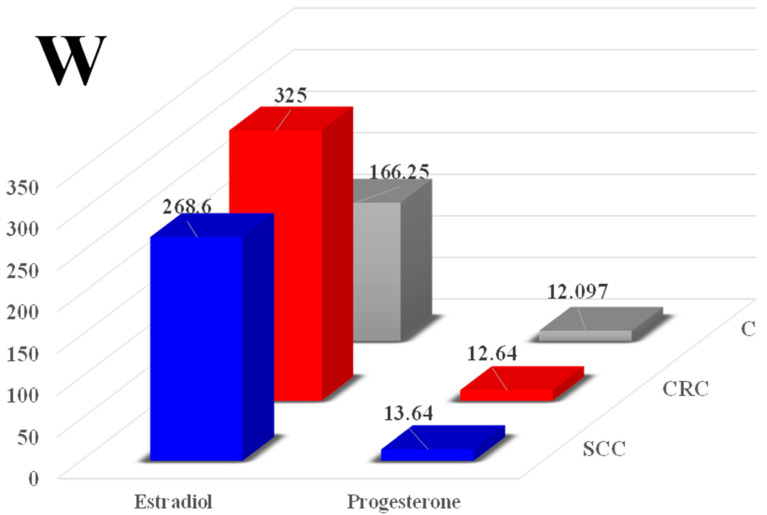
The effect of zearalenone on the mean blood levels of estradiol (E_2_, pg/mL) and progesterone (P_4_, ng/mL) in the women in groups SCC (sigmoid colorectal cancer), CRC (colorectal cancer), and C (control). Significant differences were not noted.

**Figure 4 toxins-16-00015-f004:**
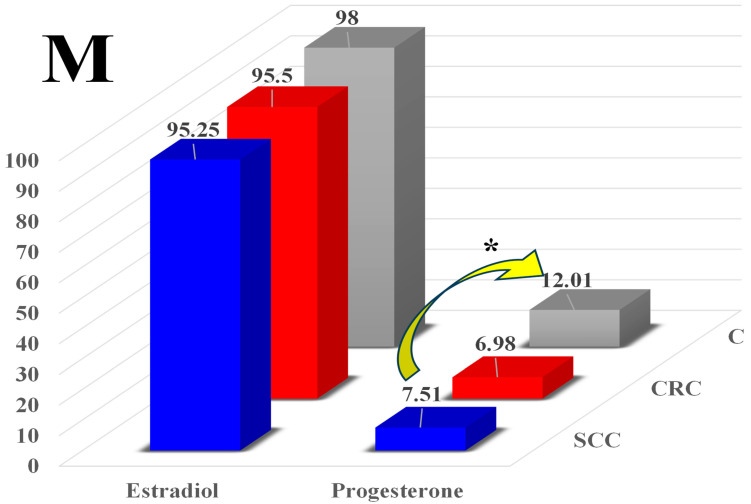
The effect of zearalenone on the average blood concentration of estradiol (E_2_, pg/mL) and progesterone (P_4_, ng/mL) in the men in groups SCC (sigmoid colorectal cancer), CRC (colorectal cancer), and C (control). Differences were regarded as statistically significant at * *p* ≤ 0.05.

**Figure 5 toxins-16-00015-f005:**
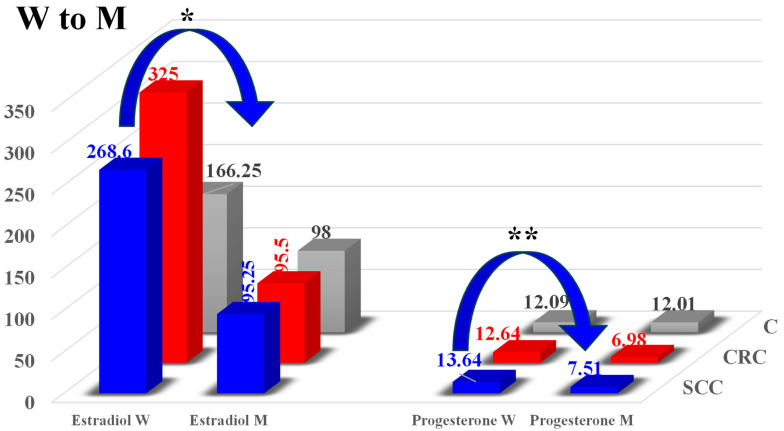
Mean concentrations of estradiol (E_2_, pg/mL) and progesterone (P_4_, ng/mL) in the peripheral blood of the women and men in groups SCC (sigmoid colorectal cancer), CRC (colorectal cancer), and C (control). Differences were regarded as statistically significant at * *p* ≤ 0.05 and ** *p* ≤ 0.01.

**Table 1 toxins-16-00015-t001:** Mean age of female and male patients diagnosed with sigmoid colorectal cancer (SCC) and colorectal cancer (CRC) who participated in a cohort study during natural zearalenone (ZEN) mycotoxicosis.

Groups	Women	Men
SCC	65.8	70.2
CRC	54.3	70.5
C	61.5	68.3

**Table 2 toxins-16-00015-t002:** The profile of the gradient elution.

Time (min)	Phase A	Phase B
0.00	80.0%	20.0%
1.00	80.0%	20.0%
4.00	0.0%	100.0%
7.00	0.0%	100.0%
7.01	80.0%	20.0%
15.00	80.0%	20.0%

**Table 3 toxins-16-00015-t003:** Optimized parameters in the MRM mode and retention times.

	Precursor	Product Ion	Fragmentor (V)	Collision Energy (V)	Retention Time	Polarity
ZEN	319.2	301.1	70	10	5.78	Positive
ZEN	319.2	283.1	70	10	5.78	Positive
α-ZEL	321.2	303.1	65	5	5.43	Positive
α-ZEL	321.2	285.1	65	5	5.43	Positive
β-ZEL	321.2	303.1	65	5	5.21	Positive
β-ZEL	321.2	285.1	65	5	5.21	Positive

**Table 4 toxins-16-00015-t004:** LOD, LOQ, and linearity for ZEN, α-ZEL, and β-ZEL.

	LOD (ng mL^−1^)	LOQ (ng mL^−1^)	Linearity (%R^2^)
ZEN	0.01	0.1	0.999
α-ZEL	0.1	0.9	0.997
β-ZEL	0.1	1	0.993

## Data Availability

Data are contained within the article.

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
