# Peer review of "A Cohort Study Investigating Zearalenone Concentrations and Selected Steroid Levels in Patients with Sigmoid Colorectal Cancer or Colorectal Cancer"

_toxins, 2023, doi:10.3390/toxins16010015_

Round 1

Reviewer 1 Report

Comments and Suggestions for Authors

The paper presents very well the aspects related to hyperestrogenism induced by ZEN, the fact that no metabolic products were identified in the blood of the patients does not mean that they did not produce lesions and were eliminated from the body, which would be interesting to investigate further. 

However, in the chapter material and methods, subchapter 4.3.1. Chromatographic Quantification of ZEN and its Metabolites,  has the chromatographic method been validated? The validation parameters must be described: values for LOD and LOQ especially obtained in the matrix, data related to the matrix effect, results for recovery (at what concentrations was spiking done?), repeatability, at what concentrations were the calibration curves done?

Comments on the Quality of English Language

Minor editing of English language required

Author Response

Dear Reviewer,

Thank you very much for your time and careful reading of the manuscript. We greatly appreciate your comments and believe that the revision of the manuscript according to your suggestions will enhance its quality and make it acceptable for the publication.

All corrections introduced in the manuscript are highlighted in blue.

However, in the chapter material and methods, subchapter 4.3.1. Chromatographic Quantification of ZEN and its Metabolites,  has the chromatographic method been validated? The validation parameters must be described: values for LOD and LOQ especially obtained in the matrix, data related to the matrix effect, results for recovery (at what concentrations was spiking done?), repeatability, at what concentrations were the calibration curves done

Dear Reviewer – all comments and suggestions of the Reviewer have been included in the text in subsection 4.3.1. Chromatographic Quantification of ZEN and its Metabolites – Table 4 and subtopic "Mass Spectrometric Conditions".

We hope that in the present form the manuscript fulfills your expectations and can be accepted for the publication.

Reviewer 2 Report

Comments and Suggestions for Authors

This is a very interesting study that raises awareness of the development of monitoring perspectives for the ZEA.

Some comments:

-        It also affects digestive [6,8], immune [8,9], and nervous systems [10]. Please, refer also to the oxidative stress imbalance (es: doi: 10.1016/j.ecoenv.2023.114571; doi: 10.3390/antiox12091748)

-        The authors' previous studies have shown that”… In general, its better to be impersonal.

-        Be careful to the abbreviations in all the text. Please, cite them in full the first time they appear in the text.

-        If you state Clinical manifestations of ZEN mycotoxicosis were not noted during patient hospitalization., why do you talk about ZEN mycotoxicosis in the whole text?

-        Analysis of the limitations of the study, will help to improve the quality of the manuscript.

Author Response

Dear Reviewer,

First of all, we would like to express our gratitude for reading and suggesting improvements to the manuscript. All of the points you raised were carefully analysed and addressed. Below please find detailed replies to your comments.

All corrections introduced in the manuscript are highlighted in green.

-        “It also affects digestive [6,8], immune [8,9], and nervous systems [10]”. Please, refer also to the oxidative stress imbalance (es: doi: 10.1016/j.ecoenv.2023.114571; doi: 10.3390/antiox12091748)

The proposals were cited. However, since oxidative stress is not any system, system or tissue, but is a normal phenomenon in the bodies of animals and humans, we have some suggestions. It should be borne in mind that it is used to describe the states of the body when the balance between free radicals and oxidative processes is disturbed. Due to this type of interpretation, we suggest citing the indicated works in the Discussion section rather than the Introduction section.

-        “The authors' previous studies have shown that”… In general, it’s better to be impersonal.

Good point - we have made the appropriate correction.

-        Be careful to the abbreviations in all the text. Please, cite them in full the first time they appear in the text.

Odpowiednie poprawki zostały naniesione w tekście publikacji.

-        If you state “Clinical manifestations of ZEN mycotoxicosis were not noted during patient hospitalization.”, why do you talk about ZEN mycotoxicosis in the whole text?

We use this term because the term zearalenone mycotoxicosis may cause clinical symptoms, but in this particular case it was mycotoxicosis, but subclinical, i.e. asymptomatic, which does not cause clinical symptoms.

-        Analysis of the limitations of the study, will help to improve the quality of the manuscript.

We don't know if we understood the reviewer correctly. We have more other research results (limitations of the study) from these patients, but we did not include them in this publication because it would greatly enlarge and deepen the scope of the discussion.

We hope that in the present form the manuscript fulfills your expectations and can be accepted for the publication.

Reviewer 3 Report

Comments and Suggestions for Authors

The manuscript reports a cohort study about zearalenone concentrations, estradiol, and progesterone levels in patients with sigmoid colorectal cancer or colorectal cancer. In my view, this topic is very meaningful. This work demonstrated novelty in its conceptualization, rigorous logic, clear composition, and comprehensive data and thus can attract much attention and expand the scope of readers. However, there are major issues to be addressed before further consideration.

1. The abstract should be rewritten according to the structure of purpose, method, result, and conclusion, and please add patient selection time and basis for grouping.

2. Introduction: Some sentences need to cite references, such as paragraph 2...

3. The aim of this study is not very clear to me. The aim of the manuscript should be explained in a better way in the introduction part.

4. Materials and Methods: Please add information for each group of patients, such as age, weight, BMI, lifestyle (drinking and smoking), pathological features, and so on.....

5. Please check and add the manufacturer and model of the instrument involved in the experiment

6.4.2. Blood Sampling: Please add the centrifugal radius.

7. Whether patient clinical data affect zearalenone concentrations, estradiol, and progesterone levels?

8. I recommend that the relation between the concentrations of zearalenone and steroid hormones be determined based on Cox or Logistic regression analysis.

Comments on the Quality of English Language

Minor editing of English language required

Author Response

Dear Reviewer,

Thank you very much for your time and careful reading of the manuscript. We greatly appreciate your comments and believe that the revision of the manuscript according to your suggestions will enhance its quality and make it acceptable for the publication.

All corrections introduced in the manuscript are highlighted in yellow.

  1. The abstract should be rewritten according to the structure of purpose, method, result, and conclusion, and please add patient selection time and basis for grouping.

The Summary chapter was edited and supplemented in accordance with the Reviewer's suggestions.

  1. Introduction: Some sentences need to cite references, such as paragraph 2...

This is our oversight and has been corrected.

  1. The aim of this study is not very clear to me. The aim of the manuscript should be explained in a better way in the introduction part.

Dear Reviewer – we have corrected the penultimate paragraph in the Introduction chapter and reworded the aim of the work. I hope it's clearer than before.

  1. Materials and Methods: Please add information for each group of patients, such as age, weight, BMI, lifestyle (drinking and smoking), pathological features, and so on.....

During ZEN exposure and from the point of view of steroid hormones, the only relevant information is the age of the patients. The others are irrelevant. With regard to note number 7 and answering the question asked in this section, we propose not to introduce any other "unnecessary" data about patients.

  1. Please check and add the manufacturer and model of the instrument involved in the experiment

Dear Reviewer, for other instruments, models and manufacturers are provided everywhere. We didn't just do this with the centrifuge

6.4.2. Blood Sampling: Please add the centrifugal radius.

According to the Reviewer's suggestion, the missing data were supplemented directly in the text.

  1. Do patient clinical data affect zearalenone concentrations, estradiol, and progesterone levels?

Clinical data (except age) do not influence the concentrations of ZEN, E2 and P4, but exposure to ZEN does influence the levels of these hormones.

  1. I recommend that the relation between the concentrations of zearalenone and steroid hormones be determined based on Cox or Logistic regression analysis.

We propose to leave the presented correlation between ZEN concentrations and steroid hormones in the formula proposed by us. Why – because it was not the aim of our research to determine predictive results for the existing clinical situation. As well, Cox regression analysis creates a predictive model for the data, but in a specific timeline to the event. In our case, we took blood samples only once on the day the patient was admitted to the ward.

We hope that in the present form the manuscript fulfills your expectations and can be accepted for the publication.

Reviewer 4 Report

Comments and Suggestions for Authors

1.      The information of group C should be written in the caption of the Figure 1.

2.      In table1, the mean age in the text cannot be found. So, the mean age of each group should be added for a more intuitive presentation of the data.

3.      Adding the standard deviation in Figure 2 may help to understand why there is no significant difference between groups as the mean of each group seems statistically significant. This situation also appears in Figure 4.

4.      ZEN has been reported to be reproductive toxic, so the reasons for choosing estradiol and progesterone as the detection indicator should be elaborated. Whether these two hormones have been reported to play a role in intestinal diseases?

5.      This study detected ZEN in the peripheral blood of patients with intestinal cancer and detected abnormal hormone secretion, but did not discuss the relationship between intestinal disease and secretion disorders induced by ZEN. This should be further discussed in the discussion part.

6.      Further detection of the mechanism between hormone levels and intestinal cancer at the cellular level may further explain the problem.

Comments on the Quality of English Language

Minor editing of English language required

Author Response

Dear Reviewer,

First of all, we would like to express our gratitude for reading and suggesting improvements to the manuscript. All of the points you raised were carefully analysed and addressed. Below please find detailed replies to your comments.

All corrections introduced in the manuscript are highlighted in violet.

  1. The information of group C should be written in the caption of the Figure 1.

Information about Group C is included in the caption Figure 1.

  1. In table1, the mean age in the text cannot be found. So, the mean age of each group should be added for a more intuitive presentation of the data.

Dear Reviewer, directly below the Table 1 there is an entire paragraph devoted to the age of patients.

  1. Adding the standard deviation in Figure 2 may help to understand why there is no significant difference between groups as the mean of each group seems statistically significant. This situation also appears in Figure 4.

Dear Reviewer – the magnitude of the standard deviations in Figure 2 was described by us in the text immediately before Figure 2. They are so large that introducing them would make the entire drawing illegible. Therefore, we propose to leave the notation on standard deviations similar to that in the paragraph preceding Figure 2 and introduced in the paragraphs describing Figure 4.

  1. ZEN has been reported to be reproductive toxic, so the reasons for choosing estradiol and progesterone as the detection indicator should be elaborated. Whether these two hormones have been reported to play a role in intestinal diseases?

Dear Reviewer – this is a very correct statement and observation, but it is very difficult to document that steroid hormones can be biomarkers of specific diseases of the reproductive system. This is due to the fact of multifactorial etiologies that initiate these disease states. On the other hand, in order to achieve this at the level of the digestive system, especially its distal section, it is even more demanding. Recently, we have published a paper on the effect of ZEN (mycoestrogen) on the activity of estrogen receptors in the final intestinal tract in pigs. We "broke through" for four years. Anyway, thank you for these kinds of insights.

  1. This study detected ZEN in the peripheral blood of patients with intestinal cancer and detected abnormal hormone secretion, but did not discuss the relationship between intestinal disease and secretion disorders induced by ZEN. This should be further discussed in the discussion part.

Due to the multidirectional activity of ZEN in the whole mammalian body, we have presented in a very condensed way in the Discussion chapter, the essence of ZEN activity in the gastrointestinal tract, especially in the terminal sections of the large intestine, as well as interactions with E2 and P4. This interaction involves inhibiting or inhibiting the process of steroid genesis. As a result, E2 is missing. It is replaced by ZEN or its metabolites that enhance the expression of ERs. As a result, the demand for P4 decreases. Recent research shows that this is accompanied by a decrease in the percentage of the thyroid hormone fT4, which together predisposes to weight gain.

The presented tests were performed from the point of view of three factors – ZEN, E2 and P4 – once at a specific time (12 to 24 hours from the moment of admission to the hospital) due to the very short half-life of ZEN. On the other hand, there are no scientific studies, or even it is impossible to extrapolate on the relationship between intestinal cancer and ZEN. This is the first attempt of its kind to document this kind of fact that it is happening at all, which may allow for more detailed research in the near future.

This is not easy because we do not even know the dose of exposure that causes pathological conditions. Patients come into contact with this undesirable substance that is ZEN in a natural way - that is for sure. As such, we would ask to be excused from discussing this issue more broadly, as this work would become a review paper on ZEN rather than a research paper.

  1. Further detection of the mechanism between hormone levels and intestinal cancer at the cellular level may further explain the problem.

This statement confirms our suggestions in Note 5. Thank you very much, because the proposed suggestion for research at the cellular level is a positive stimulus to our scientific goals.

We hope that in the present form the manuscript fulfills your expectations and can be accepted for the publication.

Round 2

Reviewer 1 Report

Comments and Suggestions for Authors

I have no other comments, the paper could be published, the authors have added almost all the requested amendments 

Comments on the Quality of English Language

Minor editing of English language required

Author Response

Thank You

Reviewer 3 Report

Comments and Suggestions for Authors

This manuscript was revised well and can be published as it is.

Author Response

Thank You